# Differential Properties of Sinkhorn Approximation for Learning with Wasserstein Distance

**Giulia Luise** [1]    **Alessandro Rudi** [2]    **Massimiliano Pontil** [1,3]    **Carlo Ciliberto** [1,4]

{g.luise.16,m.pontil}@ucl.ac.uk   alessandro.rudi@inria.fr   c.ciliberto@imperial.ac.uk

[1]Department of Computer Science, University College London, London, UK.

[2]INRIA - Département d'informatique, École Normale Supérieure - PSL Research University, Paris, France.

[3]Istituto Italiano di Tecnologia, Genova, Italy.

[4]Department of Electrical and Electronic Engineering, Imperial College, London, UK.

## Abstract

Applications of optimal transport have recently gained remarkable attention as a result of the computational advantages of entropic regularization. However, in most situations the Sinkhorn approximation to the Wasserstein distance is replaced by a regularized version that is less accurate but easy to differentiate. In this work we characterize the differential properties of the original Sinkhorn approximation, proving that it enjoys the same smoothness of its regularized version and we explicitly provide an efficient algorithm to compute its gradient. We show that this result benefits both theory and applications: on one hand, high order smoothness confers statistical guarantees to learning with Wasserstein approximations. On the other hand, the gradient formula is used to efficiently solve learning and optimization problems in practice. Promising preliminary experiments complement our analysis.

## 1   Introduction

Applications of optimal transport have been gaining increasing attention in machine learning. This success is mainly due to the recent introduction of the Sinkhorn distance [1, 2], which offers an efficient alternative to the heavy cost of evaluating the Wasserstein distance directly. The computational advantages have motivated recent applications in optimization and learning over the space of probability distributions, where the Wasserstein distance is a natural metric. However, in these settings adopting the Sinkhorn approximation requires solving a further optimization problem *with respect to* the corresponding approximation function rather than only evaluating it in a point. This consists in a bi-level problem [3] for which it is challenging to derive an optimization approach[4]. As a consequence, a regularized version of the Sinkhorn approximation is usually considered [5, 6, 7, 8, 9], for which it is possible to efficiently compute a gradient and thus employ it in first-order optimization methods [8]. More recently, also efficient automatic differentiation strategies have been proposed [10], with applications ranging from dictionary learning [11] to GANs [4] and discriminat analysis [12]. A natural question is whether the easier tractability of this regularization is paid in terms of accuracy. Indeed, while as a direct consequence of [13] it can be shown that the original Sinkhorn approach provides a sharp approximation to the Wasserstein distance [13], the same is not guaranteed for its regularized version.

In this work we recall both theoretically and empirically that in optimization problems the original Sinkhorn approximation is significantly more favorable than its regularized counterpart, which has been indeed noticed to have a tendency to find over-smooth solutions [14]. We take this as a motivation to study the differential properties of the sharp Sinkhorn with the goal of deriving a strategy to address optimization and learning problems over probability distributions. The principal contributions of this

work are twofold. Firstly, we show that both Sinkhorn approximations are highly smooth functions, namely $C^\infty$ functions in the interior of the simplex. Despite the comparable differential properties, sharp and regularized Sinkhorn approximations show a rather different behaviour when adopted in optimization problems such as the computation of barycenters [8]. As a by-product of the proof of the smoothness, we obtain an explicit formula to efficiently compute the gradient of the sharp Sinkhorn approximation, which proves to be viable alternative to automatic differentiation [10].

As a second main contribution, we provide a novel sound approach to the challenging problem of *learning with Sinkhorn loss*, recently considered in [6]. In particular, we leverage the smoothness of the Sinkhorn approximation to study the generalization properties of a structured prediction estimator adapted from [15] to this setting, proving consistency and finite sample bounds. We provide preliminary empirical evidence of the effectiveness of the proposed approach.

## 2 Background: Optimal Transport and Wasserstein Distance

Optimal transport theory investigates how to compare probability measures over a domain $X$. Given a distance function $\mathsf{d} : X \times X \to \mathbb{R}$ between points on $X$ (e.g. the Euclidean distance on $X = \mathbb{R}^d$), the goal of optimal transport is to "translate" (or lift) it to distances between probability distributions *over* $X$. This allows to equip the space $\mathcal{P}(X)$ of probability measures on $X$ with a metric referred to as *Wasserstein* distance, which, for any $\mu, \nu \in \mathcal{P}(X)$ and $p \geq 1$ is defined (see [16]) as

$$W_p^p(\mu, \nu) = \inf_{\boldsymbol{\pi} \in \Pi(\mu, \nu)} \int_{X \times X} \mathsf{d}^p(x, y) \, \mathsf{d}\boldsymbol{\pi}(x, y), \tag{1}$$

where $W_p^p$ denotes the $p$-th power of $W_p$ and where $\Pi(\mu, \nu)$ is the set of probability measures on the product space $X \times X$ whose marginals coincide with $\mu$ and $\nu$; namely

$$\Pi(\mu, \nu) = \{\boldsymbol{\pi} \in \mathcal{P}(X \times X) \text{ such that } \mathsf{P}_1 \# \boldsymbol{\pi} = \mu, \ \mathsf{P}_2 \# \boldsymbol{\pi} = \nu\}, \tag{2}$$

with $\mathsf{P}_i(x_1, x_2) = x_i$ the projection operators for $i = 1, 2$ and $\mathsf{P}_i \# \boldsymbol{\pi}$ the push-forward of $\boldsymbol{\pi}$ [16].

**Wasserstein distance on discrete measures.** In the following we focus on measures with discrete support. In particular, we consider distributions $\mu, \nu \in \mathcal{P}(X)$ that can be written as linear combinations $\mu = \sum_{i=1}^{n} a_i \delta_{x_i}$ and $\nu = \sum_{j=1}^{m} b_j \delta_{y_j}$ of Dirac's deltas centred at a finite number $n$ and $m$ of points $(x_i)_{i=1}^{n}$ and $(y_j)_{j=1}^{m}$ in $X$. In order for $\mu$ and $\nu$ to be probabilities, the vector weights $a = (a_1, \ldots, a_n)^\top \in \Delta_n$ and $b = (b_1, \ldots, b_m)^\top \in \Delta_m$ must belong respectively to the $n$ and $m$-dimensional simplex, defined as

$$\Delta_n = \left\{ p \in \mathbb{R}_+^n \ \middle| \ p^\top \mathbb{1}_n = 1 \right\} \tag{3}$$

where $\mathbb{R}_+^n$ is the set of vectors $p \in \mathbb{R}^n$ with non-negative entries and $\mathbb{1}_n \in \mathbb{R}^n$ denotes the vector of all ones, so that $p^\top \mathbb{1}_n = \sum_{i=1}^{n} p_i$ for any $p \in \mathbb{R}^n$. In this setting, the evaluation of the Wasserstein distance corresponds to solving a network flow problem [17] in terms of the weight vectors $a$ and $b$

$$W_p^p(\mu, \nu) = \min_{T \in \Pi(a, b)} \langle T, M \rangle \tag{4}$$

where $M \in \mathbb{R}^{n \times m}$ is the *cost matrix* with entries $M_{ij} = \mathsf{d}(x_i, y_j)^p$, $\langle T, M \rangle$ is the Frobenius product $\mathrm{Tr}(T^\top M)$ and $\Pi(a, b)$ denotes the *transportation polytope*

$$\Pi(a, b) = \{T \in \mathbb{R}_+^{n \times m} : \ T\mathbb{1}_m = a, \ T^\top \mathbb{1}_n = b\}, \tag{5}$$

which specializes $\Pi(\mu, \nu)$ in Eq. (2) to this setting and contains all possible joint probabilities with marginals "corresponding" to $a, b$. In the following, with some abuse of notation, we will denote by $W_p(a, b)$ the Wasserstein distance between the two discrete measures $\mu$ and $\nu$ with corresponding weight vectors $a$ and $b$.

**An Efficient Approximation of the Wasserstein Distance.** Solving the optimization in Eq. (4) is computationally very expensive [1]. To overcome the issue, the following regularized version of the problem is considered,

$$\widetilde{\mathsf{S}}_\lambda(a, b) = \min_{T \in \Pi(a, b)} \langle T, M \rangle - \frac{1}{\lambda} h(T) \qquad \text{with} \qquad h(T) = -\sum_{i, j=1}^{n, m} T_{ij}(\log T_{ij} - 1) \tag{6}$$

where $\lambda > 0$ is a regularization parameter. Indeed, as observed in [1], the addition of the entropy $h$ makes the problem significantly more amenable to computations. In particular, the optimization in Eq. (6) can be solved efficiently via Sinkhorn's matrix scaling algorithm [18]. We refer to the function $\widetilde{S}_\lambda$ as the *regularized Sinkhorn*.

In contrast to the Wasserstein distance, the regularized Sinkhorn is differentiable (actually smooth, as we show in this work in Thm. 2) and hence particularly appealing for practical applications where the goal is to solve a minimization over probability spaces. Indeed, this approximation has been recently used with success in settings related to *barycenter estimation* [8, 9, 19], supervised learning [6] and dictionary learning [7].

## 3   Motivation: a Better Approximation of the Wasserstein Distance

The computational benefit provided by the regularized Sinkhorn is paid in terms of the approximation with respect to the Wasserstein distance. Indeed, the entropic term in Eq. (6) perturbs the value of the original functional in Eq. (4) by a term proportional to $1/\lambda$, leading to potentially very different behaviours of the two functions (see Example 1 for an example of this effect in practice). In this sense, a natural candidate for a better approximation is

$$S_\lambda(a,b) = \langle T_\lambda, M \rangle \qquad \text{with} \qquad T_\lambda = \underset{T \in \Pi(a,b)}{\operatorname{argmin}} \ \langle T, M \rangle - \frac{1}{\lambda} h(T) \qquad (7)$$

that corresponds to eliminating the contribution of the entropic regularizer $h(T_\lambda)$ from $\widetilde{S}_\lambda$ *after* the transport plan $T_\lambda$ has been obtained. The function $S_\lambda$ was originally introduced in [1] as the Sinkhorn approximation, although recent literature on the topic has often adopted this name for the regularized version Eq. (6). To avoid confusion, in the following we will refer to $S_\lambda$ as the *sharp Sinkhorn*. Note that we will interchangeably use the notations $S_\lambda(a,b)$ and $S_\lambda(\mu,\nu)$. The absence of the term $h(T_\lambda)$ is reflected in a faster rate at approximating the Wasserstein distance.

**Proposition 1.** *Let $\lambda > 0$. For any pair of discrete measures $\mu, \nu \in \mathcal{P}(X)$ with respective weights $a \in \Delta_n$ and $b \in \Delta_m$, we have*

$$\big| \, S_\lambda(\mu,\nu) - W(\mu,\nu) \, \big| \leq c_1 \, e^{-\lambda} \qquad\qquad \big| \, \widetilde{S}_\lambda(\mu,\nu) - W(\mu,\nu) \, \big| \leq c_2 \lambda^{-1}, \qquad (8)$$

*where $c_1, c_2$ are constants independent of $\lambda$, depending on the support of $\mu$ and $\nu$.*

The proof of the exponential decay in error of $S_\lambda$ in Eq. (8) (Left) follows from [13] (Prop. 5.1), while the corresponding bound for $\widetilde{S}_\lambda$ Eq. (8) (Right) is a direct consequence of [20] (Prop. 2.1). Details are presented in the supplementary material. Prop. 1 suggests that the sharp Sinkhorn provides a more natural approximation of the Wasserstein distance. This intuition is further supported by the following discussion where we compare the behaviour of the two approximations on the problem of finding an optimal transport barycenter of probability distributions.

**Wasserstein Barycenters**. Finding the barycenter of a set of discrete probability measures $\mathcal{D} = (\mu_i)_{i=1}^N$ is a challenging problem in applied optimal transport settings [8]. The *Wasserstein barycenter* is defined as

$$\mu_W^* = \underset{\mu}{\operatorname{argmin}} \ \mathcal{B}_W(\mu, \mathcal{D}), \qquad \mathcal{B}_W(\mu, \mathcal{D}) = \sum_{i=1}^N \alpha_i \, W(\mu, \mu_i), \qquad (9)$$

namely the point $\mu_W^*$ minimizing the weighted average distance between all distributions in the set $\mathcal{D}$, with $\alpha_i$ scalar weights. Finding the Wasserstein barycenter is computationally very expensive and the typical approach is to approximate it with the barycenter $\tilde{\mu}_\lambda^*$, obtained by substituting the Wasserstein distance W with the regularized Sinkhorn $\widetilde{S}_\lambda$ in the the objective functional of Eq. (9). However, in light of the result in Prop. 1, it is natural to ask whether the corresponding baricenter $\mu_\lambda^*$ of the sharp Sinkhorn $S_\lambda$ could provide a better estimate of the Wasserstein one. While we defer a thorough empirical comparison of the two barycenters to Sec. 6, here we consider a simple scenario in which the sharp Sinkhorn can be proved to be a significantly better approximation of the Wasserstein distance.

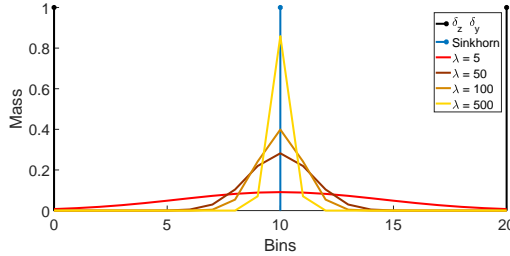

Figure 1: Comparison of the sharp (Blue) and regularized (Orange) barycenters of two Dirac's deltas (Black) centered in 0 and 20 for different values of $\lambda$.

**Example 1** (Barycenter of two Deltas). *We consider the problem of estimating the barycenter of two Dirac's deltas $\mu_1 = \delta_z, \mu_2 = \delta_y$ centered at $z = 0$ and $y = n$ with $z, y \in \mathbb{R}$ and $n$ an even integer. Let $X = \{x_0, \ldots, x_n\} \subset \mathbb{R}$ be the set of all integers between $0$ and $n$ and $M$ the cost matrix with squared Euclidean distances. Assuming uniform weights $\alpha_1 = \alpha_2$, it is well-known that the Wasserstein barycenter is the delta centered on the euclidean mean of $z$ and $y$, $\mu_W^* = \delta_{\frac{z+y}{2}}$. A direct calculation (see Appendix A) shows instead that the* regularized *Sinkhorn barycenter $\tilde{\mu}_\lambda^* = \sum_{i=0}^n a_i \delta_{x_i}$ tends to spread the mass across all $x_i \in X$, accordingly to the amount of regularization,*

$$a_i \propto e^{-\lambda((z-x_i)^2 + (y-x_i)^2)} \qquad i = 0, \ldots, n, \tag{10}$$

*behaving similarly to a (discretized) Gaussian with standard deviation of the same order of the regularization $\lambda^{-1}$. On the contrary, the* sharp *Sinkhorn barycenter equals the Wasserstein one, namely $\mu_\lambda^* = \mu_W^*$ for every $\lambda > 0$. An example of this behaviour is reported in Fig. 1.*

**Main Challenges of the Sharp Sinkhorn.** The example above, together with Prop. 1, provides a strong argument in support of adopting the sharp Sinkhorn over its regularized version. However, while the gradient of the regularized Sinkhorn can be easily computed (see [8] or Sec. 4), an explicit form for the gradient of the sharp Sinkhorn has not been considered. While approaches based on automatic differentiation have been successfully recently adopted [4, 11, 4, 12], in this work we are interested in investigating the analytic properties of the gradient of the sharp Sinkhorn, for which we provide an explicit algorithm in the following.

## 4 Differential Properties of Sinkhorn Approximations

In this section we present a proof of the smoothness of the two Sinkhorn approximations introduced above, and the explicit derivation of a formula for the gradient of $S_\lambda$. These results will be key to employ the sharp Sinkhorn in practical applications. They are obtained leveraging the Implicit Function Theorem [21] via a proof technique analogous to that in [12, 22, 23], which we outline in this section and discuss in detail in the appendix.

**Theorem 2.** *For any $\lambda > 0$, the Sinkhorn approximations $\widetilde{S}_\lambda$ and $S_\lambda : \Delta_n \times \Delta_n \to \mathbb{R}$ are $C^\infty$ in the interior of their domain.*

Thm. 2 guarantees both Sinkhorn approximations to be infinitely differentiable. In Sec. 5 this result will allow us to derive an estimator for supervised learning with Sinkhorn loss and characterize its corresponding statistical properties (i.e. universal consistency and learning rates). The proof of Thm. 2 is instrumental to derive a formula for the gradient of $S_\lambda$. We discuss here its main elements and steps while referring to the supplementary material for the complete proof.

*Sketch of the proof.* The proof of Thm. 2 hinges on the characterization of the (Lagrangian) dual problem of the regularized Sinkhorn in Eq. (6). This can be formulated (see e.g. [1]) as

$$\max_{\alpha, \beta} \mathcal{L}_{a,b}(\alpha, \beta), \qquad \mathcal{L}_{a,b}(\alpha, \beta) = \alpha^\top a + \beta^\top b - \frac{1}{\lambda} \sum_{i,j=1}^{n,m} e^{-\lambda(M_{ij} - \alpha_i - \beta_j)} \tag{11}$$

---

**Algorithm 1** Computation of $\nabla_a \mathrm{S}_\lambda(a, b)$

---

**Input:** $a \in \Delta_n$, $b \in \Delta_m$, cost matrix $M \in \mathbb{R}_+^{n,m}$, $\lambda > 0$.
   $T = \mathrm{SINKHORN}(a, b, M, \lambda)$,    $\bar{T} = T_{1:n, 1:(m-1)}$
   $L = T \odot M$,         $\bar{L} = L_{1:n, 1:(m-1)}$
   $D_1 = \mathrm{diag}(T \mathbb{1}_m)$,  $D_2 = \mathrm{diag}(\bar{T}^\top \mathbb{1}_n)^{-1}$
   $H = D_1 - \bar{T} D_2 \bar{T}^\top$,
   $\mathrm{f} = -L \mathbb{1}_m + \bar{T} D_2 \bar{L}^\top \mathbb{1}_n$
   $\mathrm{g} = H^{-1} \mathrm{f}$
**Return:** $\mathrm{g} - \mathbb{1}_n (\mathrm{g}^\top \mathbb{1}_n)$

---

with dual variables $\alpha \in \mathbb{R}^n$ and $\beta \in \mathbb{R}^m$. By Sinkhorn's scaling theorem [18], the optimal primal solution $T_\lambda$ in Eq. (7) can be obtained from the dual solution $(\alpha_*, \beta_*)$ of Eq. (11) as

$$T_\lambda = \mathrm{diag}(\mathrm{e}^{\lambda \alpha_*}) \, \mathrm{e}^{-\lambda M} \, \mathrm{diag}(\mathrm{e}^{\lambda \beta_*}), \tag{12}$$

where for any $\mathsf{v} \in \mathbb{R}^n$, the vector $\mathrm{e}^{\mathsf{v}} \in \mathbb{R}^n$ denotes the element-wise exponentiation of $\mathsf{v}$ (analogously for matrices) and $\mathrm{diag}(\mathsf{v}) \in \mathbb{R}^{n \times n}$ is the diagonal matrix with diagonal corresponding to $\mathsf{v}$.

Since both Sinkhorn approximations are smooth functions of $T_\lambda$, it is sufficient to show that $T_\lambda(a, b)$ itself is smooth as a function of $a$ and $b$. Given the characterization of Eq. (12) in terms of the dual solution, this amounts to prove that $\alpha_*(a, b)$ and $\beta_*(a, b)$ are smooth with respect to $(a, b)$, which is shown leveraging the Implicit Function Theorem [21]. $\qquad \square$

**The gradient of Sinkhorn approximations**. We now discuss how to derive the gradient of Sinkhorn approximations with respect to one of the two variables. In both cases, the dual problem introduced in Eq. (11) plays a fundamental role. In particular, as pointed out in [8], the gradient of the regularized Sinkhorn approximation can be obtained directly from the dual solution as $\nabla_a \widetilde{\mathrm{S}}_\lambda(a, b) = \alpha_*(a, b)$, for any $a \in \mathbb{R}^n$ and $b \in \mathbb{R}^m$. This characterization is possible because of well-known properties of primal and dual optimization problems [17].

The sharp Sinkhorn approximation does not have a formulation in terms of a dual problem and therefore a similar argument does not apply. Nevertheless, we show here that it is still possible to obtain its gradient in closed form in terms of the dual solution.

**Theorem 3.** *Let $M \in \mathbb{R}^{n \times m}$ be a cost matrix, $a \in \Delta_n$, $b \in \Delta_m$ and $\lambda > 0$. Let $\mathcal{L}_{a,b}(\alpha, \beta)$ be defined as in (11), with argmax in $(\alpha_*, \beta_*)$. Let $T_\lambda$ be defined as in Eq. (12). Then,*

$$\nabla_a \mathrm{S}_\lambda(a, b) = \mathrm{P}_{\mathrm{T}\Delta_n} \big( A \, L \mathbb{1}_m + B \, \bar{L}^\top \mathbb{1}_n \big) \tag{13}$$

*where $L = T_\lambda \odot M \in \mathbb{R}^{n \times m}$ is the entry-wise multiplication between $T_\lambda$ and $M$ and $\bar{L} \in \mathbb{R}^{n \times m-1}$ corresponds to $L$ with the last column removed. The terms $A \in \mathbb{R}^{n \times n}$ and $B \in \mathbb{R}^{n \times m-1}$ are*

$$[A \, B] = -\lambda D \, \big[ \nabla^2_{(\alpha, \beta)} \mathcal{L}_{a,b}(\alpha_*, \beta_*) \big]^{-1}, \tag{14}$$

*with $D = [\mathrm{I} \; \mathbf{0}]$ the matrix concatenating the $n \times n$ identity matrix $\mathrm{I}$ and the matrix $\mathbf{0} \in \mathbb{R}^{n \times m-1}$ with all entries equal to zero. The operator $\mathrm{P}_{\mathrm{T}\Delta_n}$ denotes the projection onto the tangent plane $\mathrm{T}\Delta_n = \{ x \in \mathbb{R}^n : \sum_{i=1}^n x_i = 0 \}$ to the simplex $\Delta_n$.*

The proof of Thm. 3 can be found in the supplementary material (Sec. C). The result is obtained by first noting that the gradient of $\mathrm{S}_\lambda$ is characterized (via the chain rule) in terms of the the gradients $\nabla_a \alpha_*(a, b)$ and $\nabla_a \beta_*(a, b)$ of the dual solutions. The main technical step of the proof is to show that these gradients correspond respectively to the terms $A$ and $B$ defined in Eq. (14).

To obtain the gradient of $\mathrm{S}_\lambda$ in practice, it is necessary to compute the Hessian $\nabla^2_{(\alpha, \beta)} \mathcal{L}_{a,b}(\alpha_*, \beta_*)$ of the dual functional. A direct calculation shows that this corresponds to the matrix

$$\nabla^2_{(\alpha, \beta)} \mathcal{L}(\alpha_*, \beta_*) = \begin{bmatrix} \mathrm{diag}(a) & \bar{T}_\lambda \\ \bar{T}_\lambda^\top & \mathrm{diag}(\bar{b}) \end{bmatrix}, \tag{15}$$

where $\bar{T}_\lambda$ (equivalently $\bar{b}$) corresponds to $T_\lambda$ (respectively $b$) with the last column (element) removed. See the supplementary material (Sec. C) for the details of this derivation.

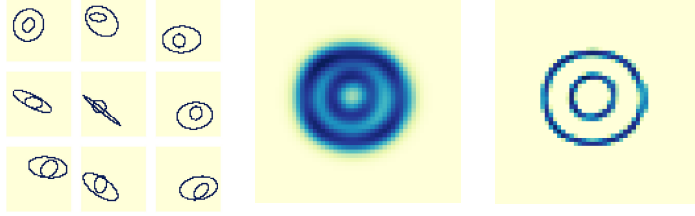

Figure 2: Nested Ellipses: (Left) Sample input data. (Middle) Regularized (Right) sharp Sinkhorn barycenters.

From the discussion above, it follows that the gradient of $S_\lambda$ can be obtained in closed form in terms of the transport plan $T_\lambda$. Alg. 1 reports an efficient approach to perform this operation. The algorithm can be derived by simple algebraic manipulation of Eq. (13), given the characterization of the Hessian in Eq. (15). We refer to the supplementary material for the detailed derivation of the algorithm.

**Barycenters with the sharp Sinkhorn.** Using Alg. 1 we can now apply the accelerated gradient descent approach proposed in [8] to find barycenters with respect to the sharp Sinkhorn. Fig. 2 reports a qualitative experiment inspired by the one in [8], with the goal of comparing the two Sinkhorn barycenters. We considered 30 images of random nested ellipses on a $50 \times 50$ grid. We interpret each image as a distribution with support on pixels. The cost matrix is given by the squared Euclidean distances between pixels. Fig. 2 shows some examples images in the dataset and the corresponding barycenters of the two Sinkhorn approximations. While the barycenter $\tilde{\mu}_\lambda^*$ of $\widetilde{S}_\lambda$ suffers a blurry effect, the $S_\lambda$ barycenter $\mu_\lambda^*$ is very sharp, suggesting a better estimate of the ideal one.

**Computational considerations.** Differentiation of sharp Sinkhorn can be efficiently carried out also via Automatic Differentiation (AD) [4]. Here we comment on the computational complexity of Alg. 1 and empirically compare the computational times of our approach and AD as dimensions and number of iterations grow. Experiments were run on a Intel(R) Xeon(R) CPU E3-1240 v3 @ 3.40GHz with 16GB RAM. The implementation of this comparison is available online[1].

By leveraging the Sherman-Woodbury matrix identity, it is possible to show that the total cost of computing the gradient $\nabla_a S_\lambda(a, b)$ with $a \in \Delta_n$ and $b \in \Delta_m$ via Alg. 1 is $O(nm \min(n, m))$. In particular, assume $m \leq n$. Then, the most expensive operations are: $O(nm^2)$ for matrix multiplication and $O(m^3)$ for inverting an $m \times m$ *positive definite* matrix. Both operations have been well-studied in the numerics literature and efficient off-the-shelf implementations (BLAS, LAPACK) are available, which exploit the low-level parallel structure of modern architectures (e.g. Cholesky and triangular inversion). Therefore, even if a priori the gradient has comparable algorithmic complexity as computing the original Wasserstein, it is reasonable to expect it to be more efficient in practice.

We compared the gradient obtained with Alg. 1 and Automatic Differentiation (AD) on random histograms with different $n$ (y axis), $m$ (x axis), and reg. $\lambda = 0.02$. From left to right, we report the ratio `time(AD) / time(Alg. 1)` for $L = 10$, $L = 50$, $L = 100$ iterations. The results shown in Fig. 3 are averaged on 10 different runs. Experiments show that there exist regimes in which the gradient computed in closed form is a viable alternative to Automatic Differentiation, depending on the task. In particular, it seems that as the ratio between the supports $n$ and $m$ of the two distributions becomes more unbalanced, Alg. 1 is consistently faster than AD.

**Accuracy and approximation errors.** We conclude this discussion on computational consideration with a note on the accuracy of the method. A priori, the expression $T_\lambda = \text{diag}(e^{\lambda \alpha_*}) \, e^{-\lambda M} \, \text{diag}(e^{\lambda \beta_*})$ which is used to derive Alg. 1 holds 'at convergence', while in practise there is a limited budget (in terms of time and memory) for the computation of $T_\lambda$, i.e. limited number of iterations. In [24] a similar issue is addressed. In Fig. 4 we empirically show that plugging an approximation $T_\lambda^L$ obtained with a fixed number $L$ of iterations in the formula for the gradient allows to reach an with respect to the 'true gradient' comparable or slightly better than automatic differentiation. Errors are measured as $\ell^2$ norm of the difference between approximated gradient and 'true gradient', where the 'true gradient' is obtained via automatic differentiation setting $10^5$ as maximum number of iterations. We show how errors decrease with respect to the number of iterations in a toy example with $n = m = 2000$ and regularization $\lambda = 0.01, 0.02, 0.05$. Examples with $n >> m$ can be found in Appendix C.1.

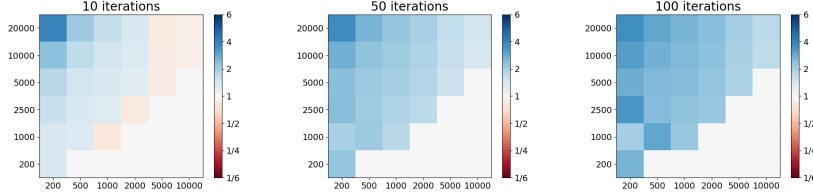

Figure 3: Ratio of `time`(AD) / `time`(Alg. 1) for 10, 50, and 100 iterations of the Sinkhorn algorithm

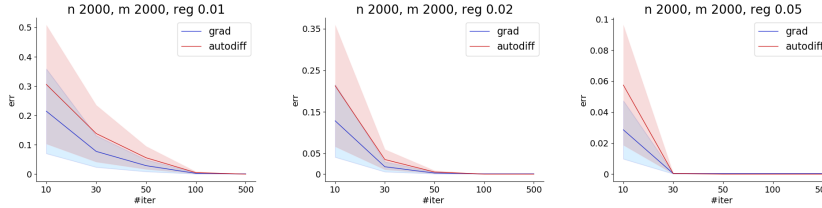

Figure 4: Accuracy of the Gradient obtained with Alg. 1 or AD with respect to the number of iterations

## 5   Learning with Sinkhorn Loss Functions

Given the characterization of smoothness for both Sinkhorn approximations, in this section we focus on a specific application: supervised learning with a Sinkhorn loss function. Indeed, the result of Thm. 2 will allow to characterize the statistical guarantees of an estimator devised for this problem in terms of its universal consistency and learning rates. Differently from [6], which adopted an empirical risk minimization approach, we address the problem in Eq. (16) from a structured prediction perspective [25] following a recent trend of works addressing the problem within the setting of statistical learning theory [15, 26, 27, 28, 29]. This will allow us to study a learning algorithm with strong theoretical guarantees that can be efficiently applied in practice.

**Problem Setting.** The problem of learning with the regularized Sinkhorn has been recently considered in [6] and can be formulated as follows. Let $\mathcal{X}$ be an input space and $\mathcal{Y} = \Delta_n$ a set of histograms. The goal is to approximate a minimizer of the *expected risk*

$$\min_{f:\mathcal{X}\to\mathcal{Y}} \mathcal{E}(f), \qquad \mathcal{E}(f) = \int_{\mathcal{X}\times\mathcal{Y}} \mathcal{S}(f(x),y)\, d\rho(x,y) \qquad (16)$$

given a finite number of training points $(x_i, y_i)_{i=1}^N$ independently sampled from the unknown distribution $\rho$ on $\mathcal{X} \times \mathcal{Y}$. The loss function $\mathcal{S} : \mathcal{Y} \times \mathcal{Y} \to \mathbb{R}$ measures prediction errors and in our setting corresponds to either $\mathrm{S}_\lambda$ or $\widetilde{\mathrm{S}}_\lambda$.

**Structured Prediction Estimator.** Given a training set $(x_i, y_i)_{i=1}^N$, we consider $\widehat{f} : \mathcal{X} \to \mathcal{Y}$ the structured prediction estimator proposed in [15], defined such that

$$\hat{f}(x) = \operatorname*{argmin}_{y\in\mathcal{Y}} \sum_{i=1}^N \alpha_i(x)\, \mathcal{S}(y, y_i) \qquad (17)$$

for any $x \in \mathcal{X}$. The weights $\alpha_i(x)$ are learned from the data and can be interpreted as scores suggesting the candidate output distribution $y$ to be close to a specific output distribution $y_i$ observed in training *according to the metric* $\mathcal{S}$. While different learning strategies can be adopted to learn the $\alpha$ scores, we consider the kernel-based approach in [15]. In particular, given a positive definite kernel $k : \mathcal{X} \times \mathcal{X} \to \mathbb{R}$ [30], we have

$$\alpha(x) = (\alpha_1(x),\dots,\alpha(x))^\top = (K + \gamma NI)^{-1} K_x \qquad (18)$$

where $\gamma > 0$ is a regularization parameter while $K \in \mathbb{R}^{N\times N}$ and $K_x \in \mathbb{R}^N$ are respectively the empirical kernel matrix with entries $K_{ij} = k(x_i, x_j)$ and the evaluation vector with entries $(K_x)_i = k(x, x_i)$, for any $i, j = 1, \dots, N$. Approaches based on *Nyström* [31] or *random features* [32] can be employed to lower the computational complexity of learning $\alpha$ from $O(n^3)$ to $O(n\sqrt{n})$ while maintaining same theoretical guarantees in the following [33, 34].

**Remark 1** (Structured Prediction and Differentiability of Sinkhorn). *The current work provides both a* theoretical *and* practical *contribution to the problem of learning with Sinkhorn approximations. On one hand, the smoothness guaranteed by Thm. 2 will allow us to characterize the generalization properties of the estimator (see below). On the other hand, Thm. 3 provides an efficient approach to* solve *the problem in Eq.* (17). *Indeed note that this optimization corresponds to solving a barycenter problem in the form of Eq.* (9).

**Generalization Properties of** $\widehat{f}$. We now characterize the theoretical properties of the estimator introduced in Eq. (17). We start by showing $\widehat{f}$ is *universally consistent*, namely that it achieves minimum expected risk as the number of training points $N$ increases. To avoid technical issues on the boundary, in the following we will require $\mathcal{Y} = \Delta_n^\epsilon$ for some $\epsilon > 0$ to be the set of points $p \in \Delta_n$ with $p_i \geq \epsilon$ for any $i = 1, \ldots, n$. The main technical step in this context is to show that for any smooth loss function on $\mathcal{Y}$, the estimator in Eq. (17) is consistent. In this sense, the characterization of smoothness in Thm. 2 is key to prove the following result, in combination with Thm. 4 in [15]. The proof can be found in the supplementary material.

**Theorem 4** (Universal Consistency). *Let $\mathcal{Y} = \Delta_n^\epsilon$, $\lambda > 0$ and $\mathcal{S}$ be either $\widetilde{\mathrm{S}}_\lambda$ or $\mathrm{S}_\lambda$. Let $k$ be a bounded continuous universal[2] kernel on $\mathcal{X}$. For any $N \in \mathbb{N}$ and any distribution $\rho$ on $\mathcal{X} \times \mathcal{Y}$ let $\widehat{f}_N : \mathcal{X} \to \mathcal{Y}$ be the estimator in Eq.* (17) *trained with $(x_i, y_i)_{i=1}^N$ points independently sampled from $\rho$ and $\gamma_N = N^{-1/4}$. Then*

$$\lim_{N \to \infty} \mathcal{E}(\widehat{f}_N) = \min_{f:\mathcal{X} \to \mathcal{Y}} \mathcal{E}(f) \quad \text{with probability } 1.$$

To our knowledge, Thm. 4 is the first result characterizing the universal consistency of an estimator minimizing an approximation to the Wasserstein distance.

**Learning Rates.** Under standard regularity conditions on the problem, our analysis also allows to prove excess risk bounds. Since these conditions are significantly technical we give an informal formulation of the theorem (see Sec. D for the rigorous statement and proof).

**Theorem 5** (Excess risk bounds - Informal). *Let $\widehat{f}_N : \mathcal{X} \to \mathcal{Y}$ be the estimator in Eq.* (17) *with $\gamma = N^{-1/2}$. Under standard regularity conditions on $\rho$ (see supplementary material),*

$$\mathcal{E}(\widehat{f}_N) - \min_{f:\mathcal{X} \to \mathcal{Y}} \mathcal{E}(f) = O(N^{-1/4})$$

*with high probability with respect to sampling of training data.*

**Remark 2.** *Recently in [36] a Sinkhorn divergence with autocorrelation terms has been proved to be a symmetric positive definite function and hence more suitable as loss function in a learning scenario. The statistical guarantees of Thm. 4 and Thm. 5 still hold true for such loss.*

We conclude this section with a note on previous work. We recall that [6] has provided the first *generalization bounds* for an estimator minimizing the regularized Sinkhorn loss. In Thm. 5 however we characterize the *excess risk bounds* of the estimator in Eq. (17). The two approaches and analysis are based on different assumptions on the problem. Therefore, a comparison of the corresponding learning rates is outside the scope of this analysis and is left for future work.

## 6 Experiments

We present here experiments comparing the two Sinkhorn approximations empirically. Optimization was performed with the accelerated gradient from [8] for $\mathrm{S}_\lambda$ and Bregman projections [9] for $\widetilde{\mathrm{S}}_\lambda$.

**Barycenters with Sinkhorn Approximations.** We compared the quality of Sinkhorn barycenters in terms of their approximation of the (ideal) Wasserstein barycenter. We considered discrete distributions on $100$ bins, corresponding to the integers from $1$ to $100$ and a squared Euclidean cost matrix $M$. We generated datasets of 10 measures each, where only $k = 1, 2, 10, 50$ (randomly chosen) consecutive bins are different from zero, with the non-zero entries sampled uniformly between $0$ and $1$ (and then normalized to sum up to 1). We empirically chose the Sinkhorn regularization parameter $\lambda$

| Improvement | 1% | Support 2% | 10% | 50% |
|---|---|---|---|---|
| $\mathcal{B}_W(\tilde{\mu}_\lambda^*) - \mathcal{B}_W(\mu_\lambda^*)$ | $14.914 \pm 0.076$ | $12.482 \pm 0.135$ | $2.736 \pm 0.569$ | $0.258 \pm 0.012$ |

Table 1: Average absolute improvement in terms of the ideal Wasserstein barycenter functional $\mathcal{B}_W$ in Eq. (9) for sharp vs regularized Sinkhorn for barycenters of random measures with sparse support.

| # Classes | Reconstruction Error (%) $S_\lambda$ | $\widetilde{S}_\lambda$ | Hell[26] | KDE [37] | Misclassification rate of the classifier (%) |
|---|---|---|---|---|---|
| 2 | $\mathbf{3.7 \pm 0.6}$ | $4.9 \pm 0.9$ | $8.0 \pm 2.4$ | $12.0 \pm 4.1$ | $0.024 \pm 0.003$ |
| 4 | $\mathbf{22.2 \pm 0.9}$ | $31.8 \pm 1.1$ | $29.2 \pm 0.8$ | $40.8 \pm 4.2$ | $0.076 \pm 0.008$ |
| 10 | $\mathbf{38.9 \pm 0.9}$ | $44.9 \pm 2.5$ | $48.3 \pm 2.4$ | $64.9 \pm 1.4$ | $0.178 \pm 0.012$ |

Table 2: Average reconstruction errors of the Sinkhorn, Hellinger, and KDE estimators on the Google QuickDraw reconstruction problem. Errors measured by a digit classifier with base misclassification reported in last column.

to be the smallest value such that the output $T_\lambda$ of the Sinkhorn algorithm would be within $10^{-6}$ from the transport polytope in 1000 iterations. Tab. 1 reports the absolute improvement of the barycenter of the sharp Sinkhorn with respect to the one obtained with the regularized Sinkhorn, averaged over 10 independent dataset generation for each support size $k$. As can be noticed, the sharp Sinkhorn consistently outperforms its regularized counterpart. The improvement is more evident for measures with sparse support and tends to reduce as the support increases. This is in line with the remark in Example 1 and the fact that the regularization term in $\widetilde{S}_\lambda$ encourages oversmoothed solutions.

**Learning with Wasserstein loss.** We evaluated the Sinkhorn approximations in an image reconstruction problem similar to the one considered in [37] for structured prediction. Given an image depicting a drawing, the goal is to learn how to reconstruct the lower half of the image (output) given the upper half (input). Similarly to [8] we interpret each (half) image as an histogram with mass corresponding to the gray levels (normalized to sum up to 1). For all experiments, according to [15], we evaluated the performance of the reconstruction in terms of the classification accuracy of an image recognition SVM classifier trained on a separate dataset. To train the structured prediction estimator in Eq. (17) we used a Gaussian kernel with bandwith $\sigma$ and regularization parameter $\gamma$ selected by cross-validation.

*Google QuickDraw.* We compared the performance of the two estimators on a challenging dataset. We selected $c = 2, 4, 10$ classes from the Google QuickDraw dataset [38] which consists in images of size $28 \times 28$ pixels. We trained the structured prediction estimators on 1000 images per class and tested on other 1000 images. We repeated these experiments 5 times, each time randomly sampling a different training and test dataset. Tab. 2 reports the reconstruction error (i.e. the classification error of the SVM classifier) over images reconstructed by the Sinkhorn estimators, the structured prediction estimator with Hellinger loss [15] and the Kernel Dependency Estimator (KDE) [37]. Last column reports the base misclassification error of the SVM classifier on the ground truth (i.e. the complete digits), providing a lower bound on the smallest possible reconstruction error. Both Sinkhorn estimators perform significantly better than their competitors (except the Hellinger distance outperforming $\widetilde{S}_\lambda$ on 4 classes). This is in line with the intuition that optimal transport metrics respect the way the mass is distributed on images [1, 8]. Moreover, it is interesting to note that the estimator of the sharp Sinkhorn provides always better reconstructions than its regularized counterpart.

# 7 Conclusions

In this paper we investigated the differential properties of Sinkhorn approximations. We proved the high order smoothness of the two functions and derived as a by-product of the proof an explicit algorithm to efficiently compute the gradient of the sharp Sinkhorn. The characterization of smoothness proved to be a key tool to study the statistical properties of the Sinkhorn approximation as loss function. In particular we considered a structured prediction estimator for which we proved universal consistency and excess risk bounds. Future work will focus on further applications and a more extensive comparison with the existing literature.

**Acknowledgments**

This work was supported in part by EPSRC Grant N. EP/P009069/1, by the European Research Council (grant SEQUOIA 724063), UK Defence Science and Technology Laboratory (Dstl) and Engineering and Physical Research Council (EPSRC) under grant EP/P009069/1. This is part of the collaboration between US DOD, UK MOD and UK EPSRC under the Multidisciplinary University Research Initiative.

## Footnotes

[1] `https://github.com/GiulsLu/OT-gradients`

[2]This is a standard assumptions for universal consistency (see [35]). Example: $k(x, x') = \mathrm{e}^{-\|x-x'\|^2/\sigma}$.

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
