[Supplementary Material · suppl_neurips.pdf]

## Supplementary Material

## A  Barycenter of Dirac Deltas

Wasserstein barycenter problems can be divided into two main classes: problems in which the support is free (and must be computed, generating a nonconvex problem [8]) and problems where the support is fixed. In some cases, the latter is the only valid choice: for instance, when the geometric domain is a space of symbols and the cost matrix $M$ contains the symbol-to-symbol dissimilarities, no extra information of the symbol space is available and the support of the barycenter will have to lie on a pre-determined set in order to be meaningful. A concrete example is the following: when dealing with histograms on words, the barycenter will optimize how to spread the mass among a set of known words that are used to build the matrix $M$, through a word2vec operation. In the following we carry out the computation of the barycenter of two Dirac deltas with regularized Sinkhorn and sharp Sinkhorn approximations, in order to prove what stated in example 1.

**Barycenter with $\widetilde{S}_\lambda$:**  Let $\mu = \delta_z$ be the Dirac delta centered at $z \in \mathbb{R}^d$ and $\nu = \delta_y$ the Dirac delta centered at $y \in \mathbb{R}^d$. We fix the set of admissible support of the barycenter $X = \{x_1, \dots, x_n\}$, where $x_i \in \mathbb{R}^d$ for any $i$. For the sake of simplicity let us assume that $X$ contains the point $(y+z)/2$. The cost matrices with mutual distances between $z$ and $X$ and $y$ and $X$ will be

$$M^z = \{\mathsf{d}(z, x_i)\}_{i=1}^n \in \mathbb{R}^n, \qquad M^y = \{\mathsf{d}(y, x_i)\}_{i=1}^n.$$

Since the support is fixed, only the masses $a = (a_1, \dots, a_n)$ of the barycenter $\tilde{\mu}_\lambda = \sum_{i=1}^n a_i \delta_{x_i}$ are to be computed. Vector $a$ is the minimizer of the following functional

$$\Delta_n \ni a \longrightarrow \mathcal{B}_{\widetilde{S}_\lambda}(a) = \frac{1}{2}\widetilde{S}_\lambda(a, \delta_z) + \frac{1}{2}\widetilde{S}_\lambda(a, \delta_y).$$

Note that since Dirac delta has mass 1 concentrated at a point, the transport polytope corresponding to $a$ and a Dirac delta is $\Pi(a, 1)$. The elements in $\Pi(a, 1)$ are those matrices $T \in \mathbb{R}^{n \times 1}$ such that $T \mathbb{1}_1 = a$ and $T^\top \mathbb{1}_n = 1$. Thus,

$$\begin{pmatrix} T_1 \\ T_2 \\ \vdots \\ T_n \end{pmatrix} (1) = \begin{pmatrix} a_1 \\ a_2 \\ \vdots \\ a_n \end{pmatrix} \tag{19}$$

which implies $T_1 = a_1, \dots, T_n = a_n$. In this case, $\Pi(a, 1)$ contains only one matrix, which coincides with $a^\top$. The distance $\widetilde{S}_\lambda(a, \delta_z)$ is given by $\langle a^\top, M^z \rangle - \frac{1}{\lambda} h(a)$ and, similarly, $\widetilde{S}_\lambda(a, \delta_y) = \langle a^\top, M^y \rangle - \frac{1}{\lambda} h(a)$. Then, the goal is to minimize

$$a \longrightarrow \frac{1}{2}\langle a, M^z \rangle + \frac{1}{2}\langle a, M^y \rangle + \frac{1}{\lambda}\sum_{i=1}^n a_i(\log a_i - 1)$$

with the constraint that $a \in \Delta_n$. The partial derivative with respect to $a_i$ is given by

$$\frac{\partial \mathcal{B}_{\widetilde{S}_\lambda}}{\partial a_i} = \frac{1}{2}(M_i^z + M_i^y) + \frac{1}{\lambda}\log a_i$$

Setting it equal to zero, it yields $a_i = e^{-\lambda(M_i^z + M_i^y)/2}$. The constraint $a \in \Delta_n$ leads to

$$a_i = \frac{e^{-\lambda(M_i^z + M_i^y)/2}}{\sum_{j=1}^n e^{-\lambda(M_j^z + M_j^y)/2}}.$$

Then the barycenter $\tilde{\mu}_\lambda^*$ has masses $(a_1, \dots, a_n)$ where each $a_i$ is strictly positive, with maximum at the entry corresponding to the point $x_i$ which realizes the minimum distance from $z$ and $y$, i.e. $(z+y)/2$. The sparsity of the initial deltas is lost.

**Barycenter with $S_\lambda$:** On the other hand, let us compute the barycenter between $\mu$ and $\nu$ with respect to the Sinkhorn approximation recalled in (7). The very same considerations on $\Pi(a, 1)$ still hold, so $\Pi(a, 1)$ contains $T = a^\top$ only. Hence, in this case the Sinkhorn barycenter functional $\mathcal{B}_{S_\lambda}$ coincides with the Wasserstein barycenter functional $\mathcal{B}_W$, since $S_\lambda(a, \delta_j) = \langle a^\top, M^j \rangle = W(a, \delta_j)$, for $j = z, y$. This trivially implies that $\mu_\lambda^* = \mu_W^*$.

# B  Proof of Proposition 1 in section 3

**Proposition 1.** *Let $\lambda > 0$. For any pair of discrete measures $\mu, \nu \in \mathcal{P}(X)$ with respective weights $a \in \Delta_n$ and $b \in \Delta_m$, we have*

$$\big| \, S_\lambda(\mu, \nu) - W(\mu, \nu) \, \big| \leq c_1 \, e^{-\lambda} \qquad\qquad \big| \, \widetilde{S}_\lambda(\mu, \nu) - W(\mu, \nu) \, \big| \leq c_2 \lambda^{-1}, \qquad (8)$$

*where $c_1, c_2$ are constants independent of $\lambda$, depending on the support of $\mu$ and $\nu$.*

*Proof.* As shown in [13](Prop.5.1), the sequence $T_\lambda$ converges to an optimal plan of W as $\lambda$ goes to infinity. More precisely,

$$T_\lambda \to T^* = \operatorname{argmax}_{T \in \Pi(a,b)} \{ h(T); \ \langle T, M \rangle = W(\mu, \nu) \}$$

exponentially fast, that is $\|T_\lambda - T^*\|_{\mathbb{R}^{nm}} \leq c \, e^{-\lambda}$. Thus,

$$|S_\lambda(\mu, \nu) - W(\mu, \nu)| = |\langle T_\lambda, M \rangle - \langle T^*, M \rangle| \leq \|T_\lambda - T^*\| \|M\| \leq c \, e^{-\lambda} \|M\| =: c_1 e^{-\lambda}.$$

As for the second part, let $T^*$ be the $\operatorname{argmax}_{T \in \Pi(a,b)} \{ h(T); \ \langle T, M \rangle = W(\mu, \nu) \}$. By optimality of $T_\lambda$ and $T^*$ for their optimization problems, it holds

$$0 \leq \langle T_\lambda, M \rangle - \langle T^*, M \rangle \leq \lambda^{-1}(h(T_\lambda) - h(T^*));$$

Indeed, since $T_\lambda$ is the optimum, it attains the minimum and hence

$$\langle T_\lambda, M \rangle - \lambda^{-1} h(T_\lambda) \leq \langle T, M \rangle - \lambda^{-1} h(T)$$

for any other $T$, including $T^*$. By definition of $\widetilde{S}_\lambda$ and W, the inequalities above can be rewritten as

$$0 \leq \widetilde{S}_\lambda(\mu, \nu) - W(\mu, \nu) \leq \lambda^{-1} h(T^*) =: c_2 \lambda^{-1}$$

which goes to 0 with speed $\lambda^{-1}$ as $\lambda$ goes to infinity. $\qquad\qquad\qquad\qquad\qquad \square$

# C  Proofs on differential properties and formula of the gradient

In this section we go over all the details of the proofs sketched in section 4.

**Theorem 2.** *For any $\lambda > 0$, the Sinkhorn approximations $\widetilde{S}_\lambda$ and $S_\lambda : \Delta_n \times \Delta_n \to \mathbb{R}$ are $C^\infty$ in the interior of their domain.*

*Proof.* Let us show the proof for $S_\lambda$ first. We organize it in three steps:
*Step 1. $S_\lambda$ is smooth when $T_\lambda$ is:* when considering histograms, $S_\lambda$ depends on its argument $a$ and $b$ through the optimal coupling $T_\lambda(a, b)$, being the cost matrix $M$ fixed. Thus, since $S_\lambda$ is a smooth function of $T_\lambda$ (being the Frobenius product of $T_\lambda$ with a constant matrix), showing that $S_\lambda$ is smooth in $a, b$ amounts to showing that $T_\lambda$ is smooth.

*Step 2. $T_\lambda$ is smooth when $(\alpha_*, \beta_*)$ is:* By Sinkhorn's scaling theorem [18], the optimal plan $T_\lambda$ is characterized as follows

$$T_\lambda = \operatorname{diag}(e^{\lambda \alpha_*}) e^{-\lambda M} \operatorname{diag}(e^{\lambda \beta_*}.) \qquad\qquad (20)$$

Being the exponential a smooth function, $T_\lambda(a, b)$ is smooth in $a$ and $b$ if the dual optima $\alpha_*(a, b)$ and $\beta_*(a, b)$ are. Our goal is then showing smoothness with respect to $a$ and $b$ of the dual optima.

*Step 3. $(\alpha_*, \beta_*)$ is smooth in $a, b$:* this is the most technical part of the proof. First of all, let us stress that one among the $n + m$ rows/columns constraints of $\Pi(a, b)$ is *redundant*: the standard dual problem recalled in Eq. (11) has an extra dual variable, and this degree of freedom is clear noticing

that if $(\alpha, \beta)$ is feasible, than the pair $(\alpha + t\mathbb{1}_n, \beta - t\mathbb{1}_m)$ is also feasible. In the following, we get rid of the redundancy removing one of the dual variables. Hence, let us set

$$\mathcal{L}(a, b; \alpha, \beta) = -\alpha^\top a - \beta^\top \bar{b} + \sum_{i,j=1}^{n,m-1} \frac{\mathrm{e}^{-\lambda(M_{ij} - \alpha_i - \beta_j)}}{\lambda},$$

where $\bar{b}$ corresponds to $b$ with the last element removed.

To avoid cumbersome notation, from now on we denote $x = (a, b)$ and $\gamma = (\alpha, \beta)$. The function $\mathcal{L}$ is smooth and strictly convex in $\gamma$: hence, for every fixed $x$ in the interior of $\Delta_n \times \Delta_n$ there exist $\gamma^*(x)$ such that $\mathcal{L}(x; \gamma^*(x)) = \min_\gamma \mathcal{L}(x; \gamma)$. We now fix $x_0$ and show that $x \mapsto \gamma^*(x)$ is $\mathrm{C}^k$ on a neighbourhood of $x_0$. Set $\Psi(x; \gamma) := \nabla_\gamma \mathcal{L}(x; \gamma)$; the smoothness of $\mathcal{L}$ ensures that $\Psi \in \mathrm{C}^k$. Fix $(x_0; \gamma_0)$ such that $\Psi(x_0; \gamma_0) = 0$. Since $\nabla_\gamma \Psi(x; \gamma) = \nabla_\gamma^2 \mathcal{L}(x; \gamma)$ and $\mathcal{L}$ is strictly convex, $\nabla_\gamma \Psi(x_0; \gamma_0)$ is invertible. Then, by the implicit function theorem, there exist a subset $U_{x_0} \subset \Delta_n \times \Delta_n$ and a function $\phi : U_{x_0} \to \Delta_n \times \Delta_n$ such that

i) $\phi(x_0) = \gamma_0$

ii) $\Psi(x, \phi(x)) = 0, \qquad \forall x \in U_{x_0}$

iii) $\phi \in \mathrm{C}^k(U_{x_0})$.

For each $x$ in $U_{x_0}$, since $\phi(x)$ is a stationary point for $\mathcal{L}$ and $\mathcal{L}$ is strictly convex, then $\phi(x) = \gamma^*(x)$, which is- recalling the notation set before- $(\alpha_*, \beta_*)$. By a standard covering argument, $(\alpha_*, \beta_*)$ is $\mathrm{C}^k$ on the interior of $\Delta_n \times \Delta_n$. As this holds true for any $k$, the optima $(\alpha_*, \beta_*)$, and hence $\mathrm{S}_\lambda$, are $\mathrm{C}^\infty$ on the interior of $\Delta_n \times \Delta_n$.

Let us now focus on the smoothness of $\widetilde{\mathrm{S}}_\lambda$. Note that when $a, b$ belong to the interior of the simplex, all components are strictly positive. From the characterization of $T_\lambda$ recalled in Eq. (20), we know $T_{\lambda ij} > 0$ for any $i, j = 1 \ldots n, m$. Then, since the logarithm is a smooth function of $T_\lambda$, the term $\lambda^{-1} h(T_\lambda)$ is smooth in $a$ and $b$. This fact combined with the first part of the proof shows the smoothness of $\widetilde{\mathrm{S}}_\lambda(a, b) = \langle T_\lambda, M \rangle - \lambda^{-1} h(T_\lambda)$. $\qquad\square$

With a similar procedure, the implicit function theorem provides a formula for the gradient of sharp Sinkhorn.

**Theorem 3.** *Let $M \in \mathbb{R}^{n \times m}$ be a cost matrix, $a \in \Delta_n$, $b \in \Delta_m$ and $\lambda > 0$. Let $\mathcal{L}_{a,b}(\alpha, \beta)$ be defined as in (11), with argmax in $(\alpha_*, \beta_*)$. Let $T_\lambda$ be defined as in Eq. (12). Then,*

$$\nabla_a \mathrm{S}_\lambda(a, b) = \mathsf{P}_{\mathrm{T}\Delta_n}\left(A\, L\mathbb{1}_m + B\, \bar{L}^\top \mathbb{1}_n\right) \tag{13}$$

*where $L = T_\lambda \odot M \in \mathbb{R}^{n \times m}$ is the entry-wise multiplication between $T_\lambda$ and $M$ and $\bar{L} \in \mathbb{R}^{n \times m-1}$ corresponds to $L$ with the last column removed. The terms $A \in \mathbb{R}^{n \times n}$ and $B \in \mathbb{R}^{n \times m-1}$ are*

$$[A\ B] = -\lambda D\, \left[\, \nabla^2_{(\alpha,\beta)} \mathcal{L}_{a,b}(\alpha_*, \beta_*)\,\right]^{-1}, \tag{14}$$

*with $D = [\mathrm{I}\ \mathbf{0}]$ the matrix concatenating the $n \times n$ identity matrix $\mathrm{I}$ and the matrix $\mathbf{0} \in \mathbb{R}^{n \times m-1}$ with all entries equal to zero. The operator $\mathsf{P}_{\mathrm{T}\Delta_n}$ denotes the projection onto the tangent plane $\mathrm{T}\Delta_n = \{x \in \mathbb{R}^n : \sum_{i=1}^n x_i = 0\}$ to the simplex $\Delta_n$.*

*Proof.* Let us adopt the same notation as in the previous proof. Since $\Psi = \nabla_{(\alpha,\beta)} \mathcal{L}$, by a direct computation, $\Psi$ can be written as

$$\Psi(a, b; \alpha, \beta) = \begin{pmatrix} a - C\mathbb{1} \\ b - C^\top \mathbb{1} \end{pmatrix},$$

where $C$ is the $n \times m - 1$ matrix given by $\mathrm{diag}(\mathrm{e}^{\lambda \alpha_*}) \mathrm{e}^{\lambda \bar{M}} \mathrm{diag}(\mathrm{e}^{\lambda \beta_*})$ and $\bar{M}$ is the matrix $M$ with $m^{th}$ column removed. In the following, we keep track of the dependence on $a$ only. Being $\Psi$ the gradient of $\mathcal{L}$, and $\gamma^*(a) = (\alpha_*(a), \beta_*(a))$ a stationary point, we have

$$\Psi(a; \gamma^*(a)) = 0. \tag{21}$$

For the sake of clarity, notice that:

i) $a \in \mathbb{R}^n$;

ii) $\mathcal{L} : \mathbb{R}^n \times \mathbb{R}^n \times \mathbb{R}^{m-1} \longrightarrow \mathbb{R}$, as we are considering it is a function of $a$, $\alpha$, $\beta$;

iii) $\Psi(a, \gamma(a)) = \nabla_{\alpha,\beta}\mathcal{L}(a, \gamma(a)) \in \mathbb{R}^{n+m-1 \times 1}$;

iv) $\alpha_* : \mathbb{R}^n \to \mathbb{R}^n$, $\beta_* : \mathbb{R}^n \to \mathbb{R}^{m-1}$, thus $\gamma^* : \mathbb{R}^n \to \mathbb{R}^n \times \mathbb{R}^{m-1}$.

Our goal is to derive $\nabla_a\gamma^*(a)$: by matrix differentiation rules [39] and Eq. (21),

$$\nabla_a\Psi(a, \gamma^*(a)) = \nabla_1\Psi(a, \gamma^*(a)) + \nabla_a\gamma^*(a)\nabla_2\Psi(a, \gamma^*(a)) = 0. \qquad (22)$$

Let us analyse each term: $\nabla_1\Psi(a, \gamma^*(a)) = [\mathbf{I}_n, \mathbf{0}_{n,m-1}]$ is $n \times n+m-1$ matrix with identity and zeros block, and $\nabla_2\Psi(a, \gamma^*(a)) = \nabla^2_{\alpha,\beta}\mathcal{L}(a, \gamma^*(a)) =: H$ is the Hessian of $\mathcal{L}$ evaluated at $(a, \gamma^*(a))$, which is a $n+m-1 \times n+m-1$ matrix. Together with Eq. (22), this yields

$$\nabla_a\gamma^*(a) = [\nabla_a\alpha_*(a), \nabla_a\beta_*(a)] = -DH^{-1}.$$

For the sake of clarity, note that $\nabla_a\alpha_*(a)$ and $\nabla_a\beta_*(a)$ contains the gradients of the components as columns, i.e.

$$\nabla_a\alpha_* = (\nabla_a\alpha_{*1}, \quad \nabla_a\alpha_{*2}, \quad \ldots, \quad \nabla_a\alpha_{*n})$$
$$\nabla_a\beta_* = (\nabla_a\beta_{*1}, \quad \nabla_a\beta_{*2}, \quad \ldots, \quad \nabla_a\beta_{*m-1}).$$

Now, since $S_\lambda(a,b) = \langle T_\lambda, M \rangle$ and $T_\lambda$ corresponds to Eq. (20) a straightforward computation shows that

$$\nabla_aS_\lambda(a,b) = \sum_{i,j=1}^{n,m}\nabla_aT_{\lambda ij}M_{ij} = \lambda\sum_{i,j=1}^{n,m}T_{\lambda ij}M_{ij}\nabla_a\alpha_{*i} + \lambda\sum_{i,j=1}^{n,m-1}T_{\lambda ij}M_{ij}\nabla_a\beta_{*j}.$$

Setting $L := T_\lambda \odot M$, then the formula above can be rewritten as

$$\nabla_aS_\lambda(a,b) = \lambda\sum_{i}^{n}\nabla_a\alpha_{*i}\sum_{j=1}^{m}L_{ij} + \lambda\sum_{j=1}^{m-1}\nabla_a\beta_{*j}\sum_{i=1}^{n}L_{ij},$$

which is exactly

$$\nabla_aS_\lambda(a,b) = \lambda(\nabla_a\alpha_*L\mathbb{1}_m + \nabla_a\beta_*\bar{L}^\top\mathbb{1}_n).$$

Since by definition, the gradient belongs to the tangent space of the domain, and $a \in \Delta_n$, we project on the tangent space of the simplex, recovering $P_{T\Delta_n}\lambda(\nabla_a\alpha_*L\mathbb{1}_m + \nabla_a\beta_*\bar{L}^\top\mathbb{1}_n)$. $\qquad\square$

## C.1 Massaging the gradient to get an algorithmic-friendly form

In the proof of theorem 3 we have derived a formula for the gradient of sharp Sinkhorn approximation. In this section we further manipulate it in order to obtain an algorithmic friendly expression that also points out some interesting bits that were hidden in the formula above. All the notation has already been introduced: from now on, we will drop the $\lambda$ and denote the optimal plan by $T$ to make the notation neater.

An explicit computation of the second derivatives of $\mathcal{L}$ with respect to $\alpha_i$ and $\beta_j$ for $i = 1, \ldots, n$ and $j = 1, \ldots, m-1$ leads to the following identity

$$H = \begin{pmatrix} \operatorname{diag}(T\mathbb{1}) & \bar{T} \\ \bar{T}^\top & \operatorname{diag}(\bar{T}^\top\mathbb{1}) \end{pmatrix} = \begin{pmatrix} \operatorname{diag}(a) & \bar{T} \\ \bar{T}^\top & \operatorname{diag}(\bar{b}) \end{pmatrix}.$$

That is, $H$ is a block matrix and each block can be expressed in terms of the plan $T$. The block structure can be exploited when it comes to compute the inverse: we have shown that the gradient of the dual potentials is given by

$$[\nabla_a\alpha_*, \nabla_a\beta_*] = -DH^{-1}, \qquad D = [\mathbf{I}_n, \mathbf{0}_{n,m-1}].$$

Now, the inverse of a block matrix is again a block matrix, say

$$H^{-1} = \begin{pmatrix} K_1 & K_2 \\ K_3 & K_4 \end{pmatrix}.$$

$$L = 10 \qquad L = 30 \qquad L = 100$$

Figure 5: Barycenter for 10, 30 and 100 iterations

Then, $[\nabla_a \alpha_*, \nabla_a \beta_*] = -[K_1, K_2]$. By the formula of the block inverse, setting

$$\mathcal{K} = \mathrm{diag}(T\mathbb{1}) - \bar{T}\mathrm{diag}(\bar{T}^\top \mathbb{1})^{-1}\bar{T}^\top,$$

the blocks $K_1$ and $K_2$ are given by

$$K_1 = \mathcal{K}^{-1}, \qquad K_2 = -\mathcal{K}^{-1}\bar{T}\mathrm{diag}(\bar{T}^\top \mathbb{1})^{-1}.$$

Note that $\mathcal{K}$ is symmetric and so its inverse. Now, we can rewrite $\lambda(\nabla_a \alpha_* L\mathbb{1}_m + \nabla_a \beta_* \bar{L}^\top \mathbb{1}_n)$, with $L = T \odot M$, as

$$\lambda\big(-\mathcal{K}^{-1}S\mathbb{1}_m + \mathcal{K}^{-1}\bar{T}\mathrm{diag}(\bar{T}^\top \mathbb{1})^{-1}\bar{L}^\top \mathbb{1}_n\big)$$

and we conclude that

$$\nabla_a \mathrm{S}_\lambda(a,b) = \lambda \cdot \mathrm{solve}(\mathcal{K}, -L\mathbb{1}_m + \bar{T}\mathrm{diag}(\bar{T}^\top \mathbb{1})^{-1}\bar{L}^\top \mathbb{1}_n).$$

**Approximation errors** In the recent work [19], it has been shown that Sinkhorn-Knopp algorithm outputs a matrix $T_\lambda$ whose distance $\|T_\lambda \mathbb{1} - a\|_1 + \|T_\lambda^\top \mathbb{1} - b\|_1$ from the transport polytope $\Pi(a,b)$ is smaller than $\epsilon$ in $O(\epsilon^{-2}\log(s/N))$ iterations, where $s = \sum_{ij} e^{-\lambda M_{ij}}$ and $N = \min_{ij} e^{-\lambda M_{ij}}$. Let us denote by $M_{\max}$ and $M_{\min}$ the maximum and minimum elements of $M$ respectively. Then,

$$\frac{s}{N} = \sum_{ij} e^{-\lambda(M_{ij} - M_{\max})} \geq e^{-\lambda(M_{\min} - M_{\max})} \geq 1.$$

This yields the lower bound

$$\log\left(\frac{s}{N}\right) \geq c\lambda$$

where $c$ is a constant independent of $\lambda$. We can then conclude that Sinkhorn-Knopp algorithm returns a matrix $T_\lambda$ such that

$$\langle T_\lambda, M \rangle \leq \mathrm{W}(a,b) + \epsilon$$

in $O(n^2\epsilon^{-2}M_{\max}^2\lambda)$. Analysing how the estimation error propagates when estimating the gradient is a relevant question which deserves its own investigation. Fig. 4 empirically shows how the approximation error with respect to the correct gradient (obtained via reaching high accuracy in Sinkhorn algorithms) decreases with the number of iterations. We report in Fig. 6 three more examples, where $n$ is significantly larger than $m$.

One may wonder whether the error with respect to the true gradient, which is considerable when the number of iterations is too small, has an actual impact in practise; to this end, we computed the barycenter of the ellipses as in Fig. 2 of the main text but with $L = 10$ and $L = 30$ iterations. The result reported in Fig. 5 shows that $L = 10$ and $L = 30$ are not enough to recover the correct barycenter.

## C.2 Discussion about differentiability of Sinkhorn approximation on the boundary

We conclude this section with a few comments on the differentiability on the boundary. As in the previous paragraph, we drop $\lambda$ in the notation $T_\lambda$.

**Claim:** Sinkhorn approximation $a \mapsto \mathrm{S}_\lambda(a,b)$ is differentiable on the boundary with the exception of the case $a = b = \delta_x$, that is when $a$ equals $b$ and only one component is nonzero.

*Proof.* The proof of this statement follows from the following considerations:
**1)** when an histogram $a$ approaches the boundary on the simplex, at least one entry, say the $i^{th}$ entry,

Figure 6: Accuracy of the Gradient obtained with Alg. 1 or AD with respect to the number of iterations

goes to zero. The corresponding dual variable $\alpha_{*i}$ goes to minus infinity: indeed, if $a_i = 0$ the function $\mathcal{L}$ depends on $\alpha_i$ only through $e^{\lambda\alpha_i}$.

**2)** When $\alpha_i$ goes to $-\infty$, the $i^{th}$ row of the matrix $T$, which is $T_{ij} = e^{-\lambda(M_{ij}-\alpha_{*i}-\beta_{*j})} = e^{\lambda\alpha_{*i}}e^{-\lambda(M_{ij}-\beta_{*j})}$ has a zero row as limit. This is a necessary condition for $T$ to be in the transportation polytope, since $T\mathbb{1} = a$ is required.

**3)** Set $\mathsf{D}_1 = \mathrm{diag}(T\mathbb{1})$, $\mathsf{D}_2 = \mathrm{diag}(\bar{T}^\top\mathbb{1})$ and $L = T \odot M$. From the previous paragraph, we know

$$\frac{1}{\lambda}\nabla_a\mathsf{S}_\lambda(a,b) = -\mathcal{K}^{-1}L\mathbb{1}_m + \mathcal{K}^{-1}\bar{T}\mathsf{D}_2^{-1}\bar{L}^\top\mathbb{1}_n, \qquad \text{with } \mathcal{K} = (\mathsf{D}_1 - \bar{T}\mathsf{D}_2^{-1}\bar{T}^\top)$$

This is equivalent to

$$\frac{1}{\lambda}\nabla_a\mathsf{S}_\lambda(a,b) = (I - A)^{-1}\mathsf{D}_1^{-1}(-L\mathbb{1}_m + \bar{T}\mathsf{D}_2^{-1}\bar{T}^\top\mathbb{1}_n), \qquad \text{where } A = \mathsf{D}_1^{-1}\bar{T}\mathsf{D}_2^{-1}\bar{T}^\top. \quad (23)$$

**4)** By $T\mathbb{1} = a$ we know that $\mathsf{D}_1^{-1}$ coincides with $\mathrm{diag}(1/a)$ and its $i^{th}$ component would not be defined when $a_i = 0$. However, since in Eq. (23) $\mathsf{D}_1^{-1}$ pre-multiplies the matrices $T \odot M$ and $T$, the issue does not arise. Indeed, in $\mathsf{D}_1^{-1}T \odot M$ (and similarly in $\mathsf{D}_1^{-1}T$), the $i^{th}$ element of the diagonal of $\mathsf{D}_1^{-1}$ multiplies the $i^{th}$ row of $T \odot M$ and hence it is involved in the products

$$\frac{1}{e^{\lambda\alpha_{*i}}\sum_j e^{-\lambda(M_{ij}-\beta_{*j})}}e^{\lambda\alpha_{*i}}e^{-\lambda(M_{ij}-\beta_{*j})}M_{ij}, \qquad j = 1\ldots m.$$

But it is clear from the expression above that the term $e^{\lambda\alpha_{*i}}$, which would be zero for $a_i = 0$, cancels out.
**5)** To show that Eq. (23) is well defined we are left to prove that $(I - A)$ is invertible. This is true if $1$ is not an eigenvalue of $A$. Since $T$ is doubly stochastic, its eigenvalues are smaller or equal than $1$ [40]. Then, singular values of $\bar{T}$ are $\leq 1$. Being $\mathsf{D}_1$ and $\mathsf{D}_2$ diagonal matrices with entries of $a$ and $\bar{b}$ on the diagonal, unless $a = b = \delta_x$ the product of eigenvalues is strictly less that $1$ and this concludes the proof. $\qquad\square$

## D   Background and proofs for section 5

We recall the main definition and tools from [15] needed to fully understand what discussed in section 5. The structured prediction estimator recalled in Eq. (17) is derived in [15] for a large class of loss functions $\mathcal{S} : \mathcal{Y} \times \mathcal{Y} \to \mathbb{R}$ that are referred to as *Structure Encoding Loss Functions* (SELF) and satisfy the following assumption:

**Definition 1** (SELF)**.** *Let $\mathcal{Y}$ be a set. A function $\mathcal{S} : \mathcal{Y} \times \mathcal{Y} \to \mathbb{R}$ is a* Structure Encoding Loss Function *(SELF) if there exists a separable Hilbert space $\mathcal{H}_\mathcal{Y}$ with inner product $\langle\cdot,\cdot\rangle_{\mathcal{H}_\mathcal{Y}}$, a continuous map $\psi : \mathcal{Y} \to \mathcal{H}_\mathcal{Y}$ and a bounded linear operator $V : \mathcal{H}_\mathcal{Y} \to \mathcal{H}_\mathcal{Y}$ such that*

$$\mathcal{S}(y,y') = \langle\psi(y), V\psi(y')\rangle_{\mathcal{H}_\mathcal{Y}} \qquad y, y' \in \mathcal{H}_\mathcal{Y}. \quad (24)$$

While in [15] it has been observed that a wide range of commonly used loss functions are SELF, no such result was known for Sinkhorn loss. This work also provides an answer to this question. Let us prove the following result first:

**Theorem 6.** *(Smooth functions are SELF) Let $\mathcal{Y}$ be a compact subset of $\mathbb{R}^n$. Any function $\mathcal{S}$ : $\mathcal{Y} \times \mathcal{Y} \to \mathbb{R}$ such that $\mathcal{S} \in C^\infty(\mathcal{Y} \times \mathcal{Y})$ is SELF.*

*Proof.* By assumption $\mathcal{S} \in C^\infty(\mathcal{Y} \times \mathcal{Y})$. Since $\mathcal{Y}$ is compact,

$$C^\infty(\mathcal{Y} \times \mathcal{Y}) = C^\infty(\mathcal{Y}) \otimes C^\infty(\mathcal{Y}) \subset H^r(\mathcal{Y}) \otimes H^r(\mathcal{Y}), \tag{25}$$

for $r > n/2$, where $H^r(\mathcal{Y})$ is the Sobolev space made of $L^2$ functions weakly differentiable $r$ times [41], and $\otimes$ denotes the topological tensor product. The Sobolev space $= H^r(\mathcal{Y})$ is a Reproducing Kernel Hilbert Space (RKHS) [42] and we denote by $k_y = k(y, \cdot) \in H^r(\mathcal{Y})$ the reproducing kernel. The product space $H^r \otimes H^r$ is also an RKHS with reproducing kernel $K((y_1, y_2), (y_1', y_2')) = k(y_1, y_1')k(y_2, y_2')$, i.e. in general $K_{y,y'} = k_y \otimes k_{y'}$. Since $\mathcal{S} \in H^r \otimes H^r$, by reproducing property there exists a function $V \in H^r \otimes H^r$ such that

$$\mathcal{S}(y, y') = \langle V, k_y \otimes k_{y'} \rangle_{H^r \otimes H^r}.$$

By the isometric isomorphism $H^r \otimes H^r \cong \text{HS}(H^r, H^r)$ [43], with $\text{HS}(H^r, H^r)$ the space of Hilbert-Schmidt operators from $H^r$ to itself, it holds

$$\mathcal{S}(y, y') = \langle V, k_y \otimes k_{y'} \rangle_{H^r \otimes H^r} = \langle V, k_y \otimes k_{y'} \rangle_{\text{HS}} = \text{Tr}(V^* k_y \otimes k_{y'}) = \langle k_{y'}, V^* k_y \rangle_{H^r}, \tag{26}$$

where $V^*$ is the adjoint operator of $V$. To meet the conditions of definition 1 it remains to show that $V^*$ and $k_y$ are bounded. But $k_y$ is bounded in $H^r$ for any $y \in \mathcal{Y}$ by definition of reproducing kernel and the operator norm $\|V^*\|$ is bounded from above by the Hilbert-Schmidt norm $\|V\|_{\text{HS}}$ which is trivially bounded since $V \in \text{HS}(H^r, H^r)$. $\qquad\square$

**Corollary 7.** *The regularized and sharp Sinkhorn losses $\widetilde{\text{S}}_\lambda$ and $\text{S}_\lambda : \Delta_n^\epsilon \times \Delta_n^\epsilon \to \mathbb{R}$ are SELF.*

*Proof.* Since $\Delta_n^\epsilon \subset \Delta_n$ is compact and $\widetilde{\text{S}}_\lambda$, $\text{S}_\lambda$ are $C^\infty$ in the interior on $\Delta_n \times \Delta_n$ by Thm. 2, a direct application of the result above shows that $\widetilde{\text{S}}_\lambda$ and $\text{S}_\lambda$ are SELF. $\qquad\square$

Summing up these elements, the proof of Thm. 4 easily follows:

**Theorem 4** (Universal Consistency). *Let $\mathcal{Y} = \Delta_n^\epsilon$, $\lambda > 0$ and $\mathcal{S}$ be either $\widetilde{\text{S}}_\lambda$ or $\text{S}_\lambda$. Let $k$ be a bounded continuous universal[3] kernel on $\mathcal{X}$. For any $N \in \mathbb{N}$ and any distribution $\rho$ on $\mathcal{X} \times \mathcal{Y}$ let $\widehat{f}_N : \mathcal{X} \to \mathcal{Y}$ be the estimator in Eq. (17) trained with $(x_i, y_i)_{i=1}^N$ points independently sampled from $\rho$ and $\gamma_N = N^{-1/4}$. Then*

$$\lim_{N \to \infty} \mathcal{E}(\widehat{f}_N) \;=\; \min_{f:\mathcal{X} \to \mathcal{Y}} \mathcal{E}(f) \quad \text{with probability } 1.$$

*Proof.* Since $\widetilde{\text{S}}_\lambda$, $\text{S}_\lambda$ are SELF function and $\Delta_n^\epsilon$ is compact, the result follows from Thm. 4 in [15]. $\qquad\square$

We conclude the section with some comments on Thm. 5 and its proof. We have shown that $\widetilde{\text{S}}_\lambda$ and $\text{S}_\lambda$ are SELF and can be written as

$$\text{S}_\lambda(y, y') = \langle k_y, V k_{y'} \rangle_{H^r(\Delta_n^\epsilon)} \tag{27}$$

with $k$ the reproducing kernel of the Sobolev space $H^r(\Delta_n^\epsilon)$. With the same notation as in [15], let us set

$$g^*(x) = \int_{\Delta_n^\epsilon} k_y \, d\rho(y \,|\, x),$$

which corresponds to a kernel embedding. Recall that $k : \mathcal{X} \times \mathcal{X} \to \mathbb{R}$ is a positive definite kernel on $\mathcal{X}$. Let us denote by $\mathcal{H}_\mathcal{X}$ the RKHS on $\mathcal{X}$ associated with the kernel $k$. Then, the "standard regularity" conditions mentioned in the statement of Thm. 5 amount to asking that $g^*$ belongs to the $\mathcal{H}_\mathcal{X} \otimes \mathcal{H}_\mathcal{Y}$. This paraphrases a regularity condition on the distribution $\rho$ itself and is a standard assumption in statistical learning theory [44, 45]. The formal statement reads as follows:

| #err | $\widetilde{\mathrm{S}}_\lambda$ | $\mathrm{S}_\lambda$ |
|---|---|---|
| $\widetilde{\mathrm{S}}_\lambda$ | 16 | 11 |
| $\mathrm{S}_\lambda$ | 1 | 6 |

Figure 7: (Right) Relative error (see text) for the Sinkhorn estimators on the digit reconstruction problem. (Left) Sample predictions for regularized (First image) and sharp Sinkhorn estimators.

**Theorem 8.** *Let $\mathcal{Y} = \Delta_n^\varepsilon$, $\lambda > 0$ and $\mathcal{S}$ be either $\widetilde{\mathrm{S}}_\lambda$ or $\mathrm{S}_\lambda$. Let $k : \mathcal{X} \times \mathcal{X} \to \mathbb{R}$ be a bounded continuous reproducing kernel on $\mathcal{X}$ and $\hat{f}_N : \mathcal{X} \to \mathcal{Y}$ the estimator in Eq.* (17) *with $N$ training points and $\gamma = N^{-1/2}$. If $g^* \in \mathcal{H}_\mathcal{X} \otimes \mathcal{H}_\mathcal{Y}$, then*

$$\mathcal{E}(f) - \min_{f:\mathcal{X}\to\mathcal{Y}} \mathcal{E}(f) \leq c\tau^2 N^{-1/4}$$

*holds with probability $1 - 8\mathrm{e}^{-\tau}$ for any $\tau > 0$, with $c$ a constant independent of $N$ and $\tau$.*

*Proof.* The proof substantially takes advantage of the fact that $\widetilde{\mathrm{S}}_\lambda$ and $\mathrm{S}_\lambda$ are SELF and inherits the generalization bounds proved in Thm. 5 in [15]. ∎

**Remark 3.** *A relevant question is whether the Wasserstein distance could be similarly framed in the setting of structured prediction. However, the argument used to address Sinkhorn approximations relies on their smoothness properties and cannot be extended to the Wasserstein distance, which is not differentiable. A completely different approach may still be successful and we will investigate this question in future work.*

# E   Experiment on MNIST

This last section is a short supplement to section 6. We present a small experiment on the MNIST dataset that has the same flavour as the experiment on GoogleQuickDraw dataset but addresses a more specific target: to evaluate better the quality of the prediction rather than the overall quality of the reconstructed image, we train the SVM classifier trained on a separate dataset made of 2000 examples of lower halves of digits $1, 2, 5, 8, 9$. Since the classifier is trained on lower halves only, we have selected a subset of digits with clearly diverse shapes, to disregard any legitimate vagueness. This means that any classification errors will be due to a poor prediction of the lower half.

We performed the reconstruction with both $\widetilde{\mathrm{S}}_\lambda$ and $\mathrm{S}_\lambda$ loss. We tested the performance of the two estimators on 100 examples. Fig. 7 reports the *performance* of the two estimators, as follows:

  i) the terms on the diagonal presents the number of misclassification of the lower half predicted with $\widetilde{\mathrm{S}}_\lambda$ and $\mathrm{S}_\lambda$ losses;

 ii) the number on the upper diagonal represents the number of errors occurred in the classification of the prediction with $\widetilde{\mathrm{S}}_\lambda$ on those examples that were correctly classified when reconstructed with $\mathrm{S}_\lambda$;

iii) conversely, the number on the lower diagonal represents the number of errors occurred in the prediction with $\mathrm{S}_\lambda$ on those examples that were correctly classified when reconstructed with $\widetilde{\mathrm{S}}_\lambda$.

To be more precise, denote by $\mathrm{L}(\widetilde{\mathrm{S}}_\lambda)$ the vector with labels predicted by the classifier when tested on the halves of digits predicted with $\widetilde{\mathrm{S}}_\lambda$ loss and analogously $\mathrm{L}(\mathrm{S}_\lambda)$ the vector with labels given by the classifier tested on the halves of images predicted with $\mathrm{S}_\lambda$ loss. Vector L is the vector with the true labels of the test set. Consider two vectors $\tilde{e}^\lambda \in \{0,1\}^{100}$ and $e^\lambda \in \{0,1\}^{100}$ defined as follows:

$$\tilde{e}_i^\lambda = \begin{cases} 0 & \text{if } L_i = \mathrm{L}(\widetilde{\mathrm{S}}_\lambda)_i \\ 1 & \text{otherwise} \end{cases} \qquad e^\lambda{}_i = \begin{cases} 0 & \text{if } L_i = \mathrm{L}(\mathrm{S}_\lambda)_i \\ 1 & \text{otherwise.} \end{cases}$$

Table in Fig. 7 corresponds to

$$\begin{pmatrix} \sum_i \tilde{e}_i^\lambda & \sum_i \tilde{e}_i^\lambda(1 - e_i^\lambda) \\ \sum_i e_i^\lambda(1 - \tilde{e}_i^\lambda) & \sum_i e_i^\lambda. \end{pmatrix}.$$

What we observed is the following: since the classifier was trained and tested on the lower halves only, the blurriness in the reconstruction performed with $\widetilde{S}_\lambda$ played a substantial role in the misclassification on digit $5$ in favour of digit $8$. On the other hand, the sharpness of the reconstruction with $S_\lambda$ is a major advantage for the correct classification.

## Footnotes

[3]This is a standard assumptions for universal consistency (see [35]). Example: $k(x, x') = e^{-\|x-x'\|^2/\sigma}$.