[Reviews · NeurIPS 2018]

Reviewer 1



Edit after rebuttal: Given the comparison with automatic differentiation in the rebuttal and the additional experiments, I am keen to change my vote to 6. I think the authors did a good job to argument and show that there is a place for the gradient formula to be used in practice; although more experimentation shall be required to support its use. --- Summary An in-depth theoretical analysis of Sinkhorn distances is given. In particular, differentiability and gradients are studied and used in simple experiments on finding Wasserstein barycenters and learning to reconstruct missing part of images. Detailed comments I like proposition 1 as it is a solid advocate of using the “sharp” Sinkhorn distance, instead of the regularized distance. I have concern about the evolution of the paper from here on. While I appreciate the technical analysis of differentiability and gradient of Sinkhorn distances, I struggle to find any clear practical utility. Differentiability: while implementing the Sinkhorn algorithm as an approximation of the Wasserstein distance, it is immediately obviously that it is about staking multiple differentiable operations (matrix multiplication, element-wise division and exponentiation at the end). In fact, several paper already noticed and exploited this property in order to differentiate through the Sinkhorn algorithm in combination with neural networks. Gradient explicit formula: in the era of deep learning and the availability of automatic differentiation in any library, I don’t see the use for that formula (which implementation will need to be debugged!). One can simply plug the solution of Sinkhorn algorithm back into the Frobenius product, as done in (7); then auto-diff that expression. Appendix: I have only checked sections A and B and they seemed sound. I am open to be convinced otherwise by the authors, in case I am missing something. But I believe my concerns disqualify the paper for publication.

Reviewer 2



The paper focuses on the Sinkhorn approximation of Wasserstein distance, which is used to compare probability distributions and finds applications e.g. in image analysis. In particular, the authors derive methods for efficiently computing the “sharp” Sinkhorn approximation, in place of its regularized version, which the authors show to be inferior to the sharp version. The authors provide theoretical analysis of the two approximations, including proof of consistency of supervised learning with with the two Sinkhorn losses. In experiments the authors demonstrate the benefits of the sharp Sinkhorn in providing better classification results. Update: The author response is convincing, especially the comparison to AD approach. A good job!

Reviewer 3



In this paper the authors advocate the use of the Sinkhorn distance over the "regularized" sinkhorn distance for computing divergence between discrete distributions. They show that the gradient of the former is better and leads to sharper results especially on barycenters. They also provide a close form expression for the gradient of the Sinkhorn distance using the implicit function theorem. Another contribution is a new generalization bound for structured prediction with Wasserstein distance. Numerical experiments are very short be show a better barycenter with the Sinkhorn distance and better reconstruction for images. The paper is nice and some important results are presented. Still it lacks a few references and the numerical experiments are very limited as discussed more in details below. + The Sinkhorn distance in equation (7) has already been used (and optimized) in the literature over the regularized version for a few years but it needs to be properly differentiated from the regularized version and the authors did a good job there. + In [1], a reference cited in the paper, the authors did the gradient computation of the Sinkhorn distance using autodiff tools (gradient propagation along the Sinkhorn iterations). Those approaches are very efficient and come with a small overhead but are not discussed at all in the paper. The Sinkhorn distance has been also derived for discriminant subspace estimation [2]. Even though autodiff was used, the implicit function theorem is used in the supplementary material of [2] to compute the gradient of a bilevel optimization problem with Sinkhorn distance wrt to a linear operator on the samples. + The discussion about the complexity of the proposed approach is a bit misleading (with vague assumtion about teh necessary number of iteration of Sinkhorn). The complexity of the proposed SInkhorn+gradient computation is O(n^3) to get a sharp gradient. In other words the authors propose an approach to compute a better gradient that regularized sinkhorn but it comes at the cost of the same complexity of the unregularized OT (that is O(n^3) for solving, the gradoients are the dual potentials also returned by a dual solver). Since one of the important property is that the proposed gradient recover the true unregularized Wasserstein barycenter, why not use the true gradeint with same complexity? Also Autodiff is O(Ln^2) where L is the number of sinkhorn iterations so unless there is a true numerical (as in precision) gain to the propose gradient autodiff will be more efficient and needs to be discussed. + The numerical experiments are interesting but a bit short. Figure 2 is nice but it should be nice to show the impact of the regularization (in computational time if not on the final result). Also what is the result when using Autodiff as in [1]? + The generalization bound for structures learning are very nice. But the numerical experiments are very limited and hard to interpret even as a proof of concept. For instance the accuracy of a classifier on the reconstructed image is used to see if the reconstruction was accurate. The authors have to be commended for using a measure of performance that is not tied to any of the divergences used . But those performances cannot really be compared or evaluated without the performance of the classifier on the original images. please add this columns to the Table in Fig. 3 so that we have an idea of the loss incurred by the reconstruction. [1] Genevay, A., Peyré, G. and Cuturi, M., 2017. Learning generative models with sinkhorn divergences. arXiv preprint arXiv:1706.00292. [2] Flamary, R., Cuturi, M., Courty, N. and Rakotomamonjy, A., 2016. Wasserstein Discriminant Analysis. arXiv preprint arXiv:1608.08063. Author feedback ============ I want to commend the authors for their detailed and impressive feedback and strongly recommend them to include this material in the final version of the paper if accepted. The new comparison and discussion of autograd make the paper much stronger. I'm still very surprised by the time for true Wasserstein, it has been known that a modern CPU bound network simplex can solve the OT problem (and give gradient estimates) for n=100 in the order of the ms so a time of 1000s for barycenter estimation suggests bad implementation.

Reviewer 4



This paper is concerned with the so-called 'sharp Sinkhorn distance', which can be understood as lying in between the Wasserstein distance (no regularization) and the vanilla Sinkhorn distance. The author's main contribution is having showed the smoothness of the sharp distance (with respect to a vector of the simplex indicating the weights of a fixed support) along with formulae for its gradient. This computation is crucial in cases when one requires to find a barycenter between measures, assuming the support is known. The authors show the sharp Sinkhorn distance outperfroms the vanilla distance an other methods in the Wasserstein barycenter problem, and in a structured prediction task as well (learning with a Wasserstein loss, which also requires the computation of a barycenter). My judgement of recommending acceptance is based on my belief the four criteria are met: The paper is clear and easy to follow. It is of high quality and contains a thorough supplement. It is original, to my understanding: I have not seen similar ideas being published elsewhere. It is relevant: they expand the tookit of machine learning by proposing a new algorithm for approximate learning of a barycenter. That being said, I still have criticism that I hope the authors will address. First, although it is commented in remark 1, I would have liked a more in-depth comparison of complexity of vanilla Sinkhorn, sharp Sinkhorn and Wasserstein. My concern here is that sharp Sinkhorn does indeed seem to be quite consuming as it involves inversion of matrices. Because of this, I tend to believe it may throw us back to the Wasserstein distance. Therefore, my 'dream' here would be seeing a more transparent table showing how much does it take in practice to compute the barycenters (something like an expanded table 1 including time) for all methods. Additionally, the authors do not seem to highlight enough (this is mentioned only in the supplementary material) that in practice none the vanilla or sharp Sinkhorn can be attained as the associated lagrange multiplied \alpha^*, \beta^* can be computed only through a fixed point iteration (Sinkhorn). When stating theoretical results this can be very subtle (see recent papers by Jonathan Weed, Phillipe Rigollet, Jason Alschuler, etc) and one should proceed with care. It would be great if the authors could at least hint on how to proceed there. Notice also a minor mistake that should be corrected: equations 6 and 7 do not make sense, there is a missing or redundant \min or \argmin sign.